# Flow Feature in Supersonic Non-Isobaric Jet near the Nozzle Edge

**Valeriy Zapryagaev ***  , **Ivan Kavun and Nikolay Kiselev**

Experimental Aerogasdynamics Laboratory, Khristianovich Institute of Theoretical and Applied Mechanics Siberian Branch of the Russian Academy of Science, 630090 Novosibirsk, Russia; i_k@list.ru (I.K.); nkiselev@itam.nsc.ru (N.K.)

**\*** Correspondence: zapr@itam.nsc.ru; Tel.: +7-383-330-77-66

**Abstract:** Using the example of studying the supersonic underexpanded jet initial section, the issue of interpreting the experimental visualization data and Pitot pressure measurement data using the results of numerical calculations (2d RANS k-$\omega$ SST) is discussed. It is shown that the gradient S-shaped feature of the gas-dynamic structure near the nozzle exit, observed in the form of a barrel shock, is a characteristic that separates the expansion and compression regions, and downstream is transformed into a barrel shock. It has been established that the reason for the observed S-shaped curvature of this feature is the axisymmetric nature of the jet flow.

**Keywords:** supersonic underexpanded jet; nozzle; Mach disk; barrel shock; shock wave

## 1. Introduction

An analysis of the experiment and numerical calculation results is necessary to understand the physical mechanism of the shock-wave flow structure formation near the nozzle exit in the supersonic non-isobaric jet. This structure is characterized by the presence of shock waves, expansion waves, and mixing layers, and is described in many publications [1–13]. In this paper, the physical mechanism for the formation of a barrel shock in the first cell of a supersonic underexpanded jet is considered. Although this issue has been studied for more than seventy years, until now, there have been various ideas about the barrel shock origin location.

For example, in [1–8], in the diagram of the initial section of an axisymmetric sonic underexpanded jet, it is shown that the barrel shock is formed directly at the nozzle exit. Figure 1a shows a flow diagram illustrating the flow structure in the first "barrel", according to those works.

On the other hand, in [9–13], it was shown that the barrel shock forms at some distance from the nozzle exit. Figure 1b shows such a flow diagram. Here, the character I denotes the point of origin of the barrel shock.

According to [2–5,10], the barrel shock forms when the characteristics of the fan of expansion waves originating at the nozzle edge (indicated as point N) undergo reflection from the concave external boundary of the jet flow in the form of a compression fan (Figure 1c). At the intersection of the reflected characteristics, there forms a barrel shock. In this case, the region I of shock origination coincides with the edge N of the nozzle. According to [1], the boundary of the jet flow is the line at which p = $p_e$, where p is the static pressure in the jet, and $p_e$ is the pressure in the ambient space.

In [12], the results of an experimental and numerical study of jets that were exhausted from a circular axisymmetric and a square nozzle were reported. It was noted that, in the flow patterns that were observed near the nozzle exit, the expansion fan gave rise to a high-gradient flow structure that was bound by the trailing edge of the fan. It should be noted that the diagram in Figure 1d differs from the diagram in Figure 1c that is shown in the same figure: for flow characteristics that emerge from the point NO to reach the opposite boundary of the jet flow, it is necessary that barrel shock is absent near the nozzle exit.

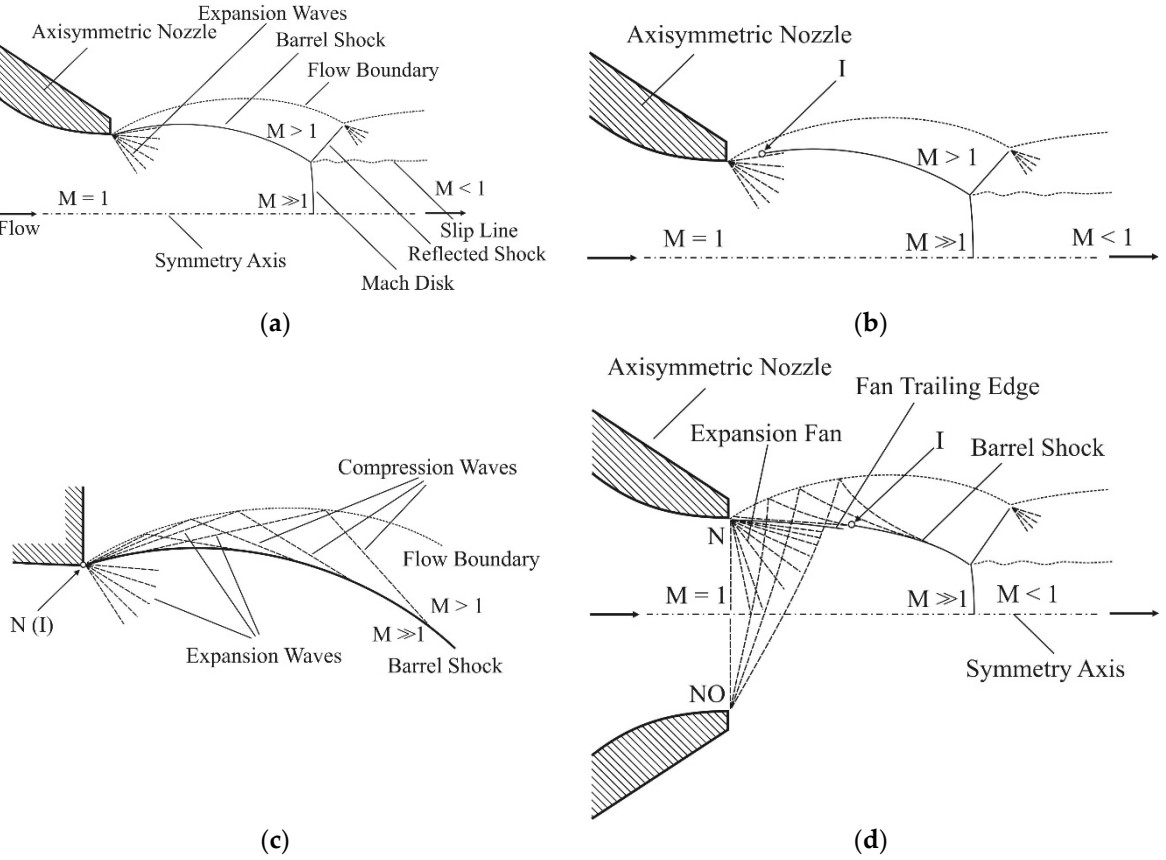

**Figure 1.** Structure of the flow in the first "barrel" of a sonic underexpanded jet: (**a**) the flow diagram, according to [1–8]; (**b**) the flow diagram, according to [9–13]; (**c**) a flow diagram illustrating the barrel-shock formation mechanism, according to [2–5,8]; (**d**) a flow diagram illustrating the barrel-shock formation mechanism, according to [10].

Conclusions about the presence or absence of a barrel shock are obtained using shadow photographs. Typical photographs (shadow and schlieren-) of the underexpanded jets that are exhausted from nozzles with various values of nozzle pressure ratio (n = $p_a/p_e$, where $p_a$ is the static pressure at the nozzle exit, and $p_e$ is the ambient pressure) are shown in Figure 2 for $M_a$ = 1, n = 3.8 (a); $M_a$ = 1, n = 8.2 (b); $M_a$ = 2, n = 2 (a); and $M_a$ = 2, n = 21 (b). Here, $M_a$ is the Mach number at the nozzle exit. The barrel shock, the reflected shock, and the Mach disk; as well as the outer boundary of the jet, or the mixing layer; and the contact discontinuity, or the slip line that emerge from the triple point of intersection of the shocks are distinctly seen. It can be noted here that in all these photographs, the observed thin dark line G, or the gradient flow feature, looks precisely like a barrel shock that reaches the nozzle edge, in the same way that it occurs in the diagram of Figure 1a.

An interesting feature of the line G, distinctly seen in both the case of jets that are exhausted out of sonic nozzles (see Figure 2a,b, where this feature is less noticeable), and in the case of jets that are exhausted from supersonic nozzles (Figure 2c) is worth noting: near the nozzle exit, the line G assumes an S-shape (region S). For a jet that is exhausted out of a nozzle with a large value of nozzle pressure ratio (see Figure 2d), this flow feature is no longer observed so distinctly.

The photograph in Figure 2c also provides a good illustration of the physical mechanism of formation of the barrel shock that is shown in the diagrams of Figure 1, with the flow characteristics being very weak nozzle shocks extending from the inner surface of the nozzle. The main difference between the photograph and the diagram of Figure 1c is that here, the characteristics do not emerge from the region N; instead, they emerge from the region that is located at the opposite edge NO of the nozzle, in the same way it is shown in

Figure 1d. Moreover, in the photograph of Figure 2d one can notice longitudinal stripes, interpreted as longitudinal vortices that are formed in the mixing layer of the jet due to Taylor–Goertler instability [6].

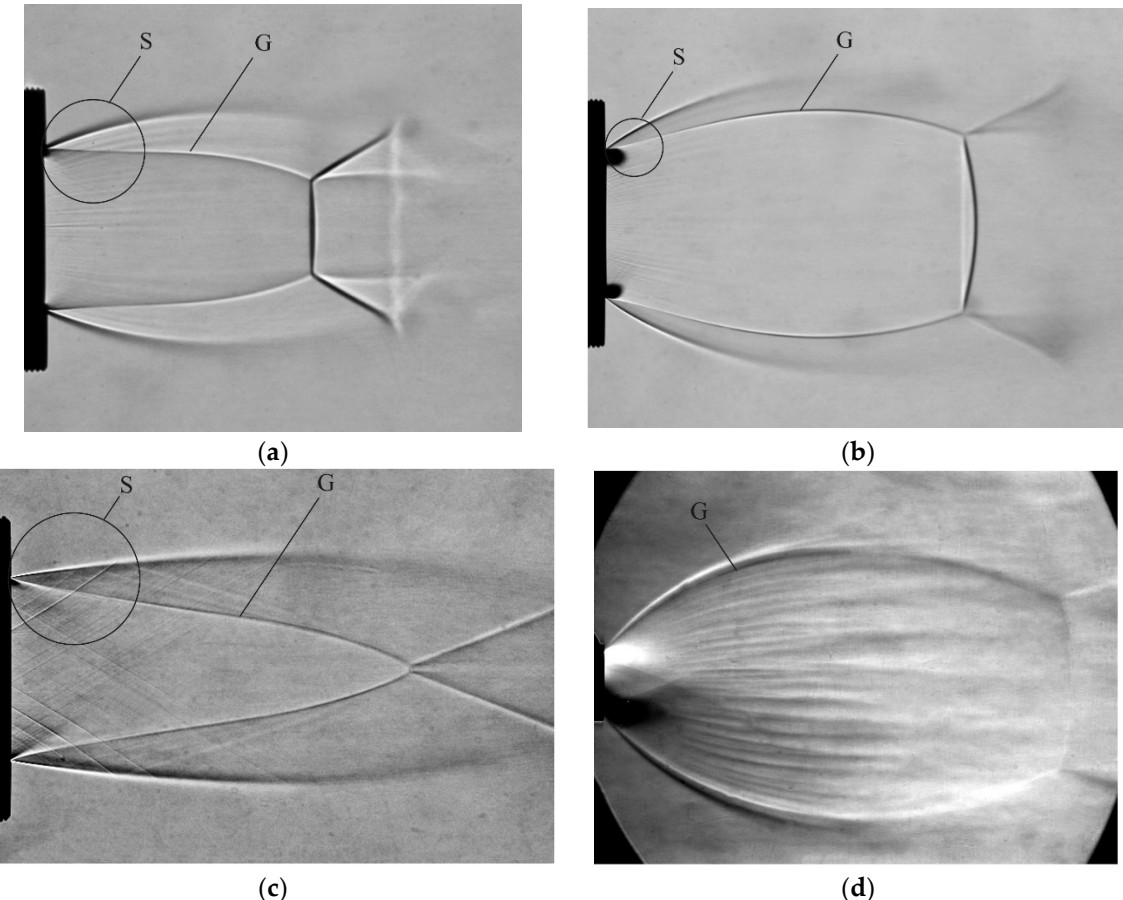

**Figure 2.** Visualization of the flow in an underexpanded jet ejected out of a sonic convergent nozzle with $M_a$ = 1.0 at flow regimes with n = 3.8 (**a**) and n = 8.2 (**b**); and out of a supersonic nozzle with $M_a$ = 2.0 at flow regimes with n = 2 (**c**) and n = 160 (**d**). Here G is the gradient flow feature, S is the region of its S-bending shape.

Such flow visualization pictures form the basis for interpreting the observed shock-wave structure of the flow of interest near the nozzle exit. At the same time, such data taken alone may appear insufficient for performing a reliable analysis. The paper analyzes the experimental and numerical data, in order to elucidate the formation mechanism of a barrel shock, the region of its occurrence, and the cause of the S-shaped curvature.

## 2. Experimental Equipment

The experiment was carried out on the jet unit of the T-326 wind tunnel of ITAM SB RAS (Figure 3). The settling chamber 1 of the jet unit was a cylindrical pipe with an inner diameter of 113 mm that was provided with a seat for mounting replaceable nozzles 2. The setup had a closed test chamber 3 whose dimensions were $1.3 \times 0.87 \times 0.93$ m³. The jet was ejected into chamber 3. Then, it entered—through the diffuser *5* of the exhaust path of the pipe—a noise suppression shaft [6,13].

The jet unit was equipped with a visualization system for gas flows and with a traversing gear that was intended for moving measuring probes.

The flow field in the jet unit was visualized using an IAB-451 shadow device through a 230-mm diameter observation window 4.

The Toepler IAB-451 device was used to obtain shadow images. The flow was visualized using a digital video camera with a high-resolution matrix and a long-focus lens. Shadow images were obtained using a horizontal Foucault knife, which allowed observation of the vertical gradient distribution of the refraction index (proportional to the gas density, Figures 2c,d and 6a). In the absence of a knife, a shadowgraph was recorded, which corresponded to the second derivative of the refraction index (see Figures 2a,b and 9b).

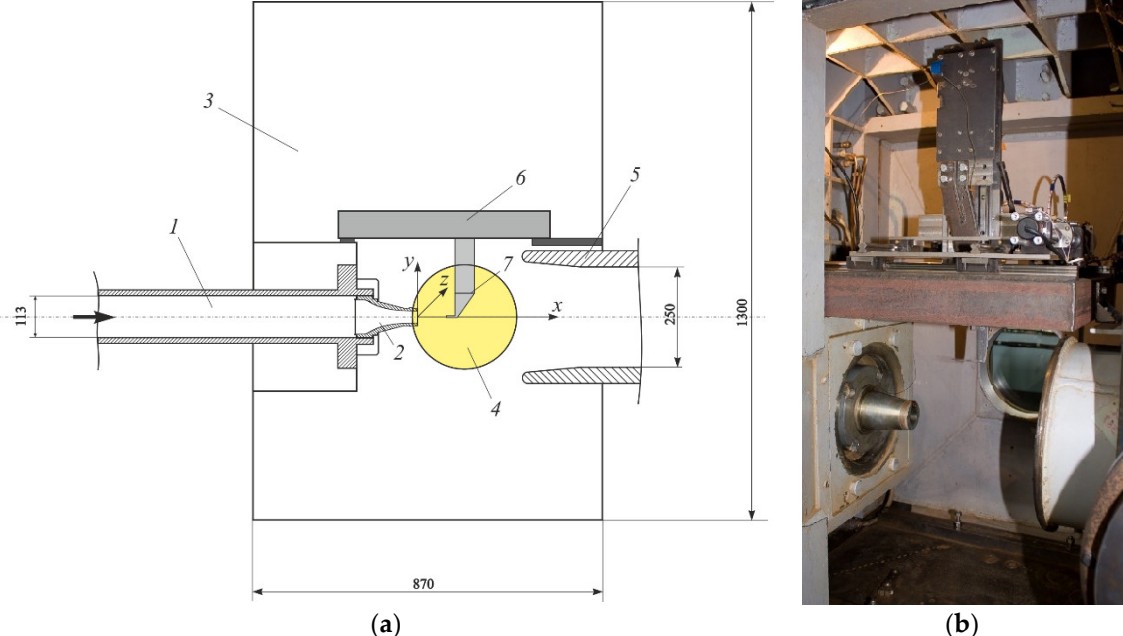

(**a**)　　　　　　　　　　　　　　　　　　　　　　　(**b**)

**Figure 3.** Scheme (**a**) and photo (**b**) of the test chamber of the jet unit of T-326 wind tunnel: 1—settling chamber; 2—nozzle; 3—test chamber; 4—observation window; 5—diffuser; 6—traversing gear intended for scanning the flow with measuring probes, 7—Pitot-tube pylon. The black bold arrow shows a flow direction.

The traversing gear 6 allowed a pylon 7 with a Pitot tube mounted on it along three coordinates (x, y, and z) to move in automatic mode. The region that could be scanned with the probe had dimensions $200 \times 200 \times 200$ mm$^3$, with the probe positioning accuracy being $\pm 20$ μm. In the experiment, a Pitot tube with an outer diameter of 0.6 mm was used.

For measuring the total pressure $p_{Pt}$ in the jet, the static pressure in the settling chamber $p_0$, and the static pressure in the test chamber $p_c$, sensors with a measurement range of up to 0.6, 25, and 0.16 MPa were used (respectively, for the pressures $p_{Pt}$, $p_0$, and $p_c$). The measurement error was 0.01% for $p_{Pt}$ and $p_c$, and 0.075% for $p_0$ (with respect to the upper limit of the measuring range).

The error in the position determination of shock waves in space depends on the measuring probe (Pitot tube) diameter that is introduced into the flow. Assuming that the maximum radial displacement of the shock wave due to the influence of a closely spaced tube is equal to half the outer diameter of the tube (0.6 mm), the error estimate in determining the position of the shocks is no more than 1% of the nozzle cut diameter.

In the experiments, an axisymmetric nozzle with the geometrical Mach number at the nozzle exit $M_a = 1.0$ was used. The nozzle exit diameter was $D_a = 30$ mm. In order to obtain a uniform stream at the outlet section of the nozzle, the contour of the nozzle was calculated using the Vitoshinsky formula [11]. The nozzle had a hydraulically smooth inner surface.

The ratio of the total pressure $p_0$ in the settling chamber to the pressure pc in the ambient space during the experiment was maintained at the level of $N_{pr} = 5$ ($N_{pr} = p_0/p_c$, $N_{pr}$ is the nozzle pressure ratio), the value of which corresponded to the jet pressure ratio n

= $p_a/p_e$ = 2.64 ($p_a$ is the pressure at the nozzle exit). The error in maintaining the value of $N_{pr}$ did not exceed 0.4%. During the experiment, the average stagnation temperature in the settling chamber was $T_0$ = 296 K, and that in the test chamber was $T_c$ = 291 K. A heater was used for maintaining the temperature $T_0$ constant. The Reynolds number that was calculated from the outlet section diameter of the nozzle was Re = $2.3 \cdot 10^6$.

## 3. Numerical Calculation

The numerical study of turbulent jet flows is currently carried out mainly using two approaches, RANS (Reynolds Averaged Navier–Stokes) and LES (Large Eddy Simulation). In the first method, the Navier–Stokes equations are averaged according to Reynolds (in compressible flows, according to Favre). The advantages are mean requirements for the computing resources (including due to the possibility of obtaining a solution in two-dimensional plane or axisymmetric formulations) and the solution satisfactory accuracy in the region of the first jet "barrel" [12,14–16].

In [14], the jet exhausting from a converging nozzle with a diameter of 29.4 mm for $N_{pr}$ = 2.5 and 4.0 was considered. The calculated data were compared with the measurements by the PIV method and the Pitot tube. The solution was obtained in the Fluent software package in an axisymmetric formulation. The k-ε RNG turbulence model was used. The grid contained 50 thousand cells. A comparison of the axial profiles showed that for $N_{pr}$ = 4 the longitudinal size of the first four "barrels" of the jet in the calculation differed from the experiment by 1–20%.

In [15], the jet exhausting from a nozzle was considered for the same pressure ratio as in the present work. The solution was obtained in the VP2/3 package in an axisymmetric formulation using the k-ω SST turbulence model in the first and second order of approximation. The grid contained 230 thousand cells. A good agreement between the calculated and experimental data is shown within the first "barrel" of the jet, especially for the solution by the second order of approximation.

In [16], the solution for the nozzle $M_a$ = 1.45 and $N_{pr}$ = 2.8–5.0 is presented. The solution was obtained in the OpenFoam package in a spatial 3d formulation by the URANS method using the k-ω SST turbulence model. The grid contained 24 million cells. A comparison of the calculated results with the schlieren photograph showed that the distance from the nozzle exit to the point of intersection of the shocks in the calculation and experiment differed by no more than 1–4% for different values of $N_{pr}$.

In the LES method, the conservation equations are obtained by filtering the unsteady Navier–Stokes equations, while large-scale eddies are calculated explicitly, and the effects that are caused by smaller eddies are modeled using a subgrid turbulence model. The method, in comparison with RANS, allows the improvement of the modeling quality of turbulent mixing layers, but it requires significantly more computational resources [5,8,17–19].

In the presented work, the numerical calculation was carried out using the ANSYS Fluent software that was intended for a numerical solution of gas-dynamics problems. The working fluid of the jet flow (air) was treated as a perfect gas obeying the Mendeleev–Clapeyron state equation. Both the dynamic viscosity μ and thermal conductivity λ were assumed to be temperature-dependent quantities, and they were calculated using the Sutherland and Eucken formulas.

The momentum and energy conservation steady equations included the turbulent (Reynolds) stress tensor [20]. The turbulent viscosity $μ_t$ that was involved in the stress tensor was calculated using a two-parameter k-ω SST model of flow turbulence [21]. The energy conservation equation took into account the gas thermal conductivity, which depended on the coefficients of molecular and turbulent viscosity of the flow.

The calculation was in two-dimensional axisymmetric and planar formulations. The computational domain (Figure 4a) contained two million tetrahedral cells with the refined mesh near the nozzle walls, its exit and the mixing layer in the vicinity of the nozzle exit (Figure 4b—a fragment of the grid near the nozzle exit is shown).

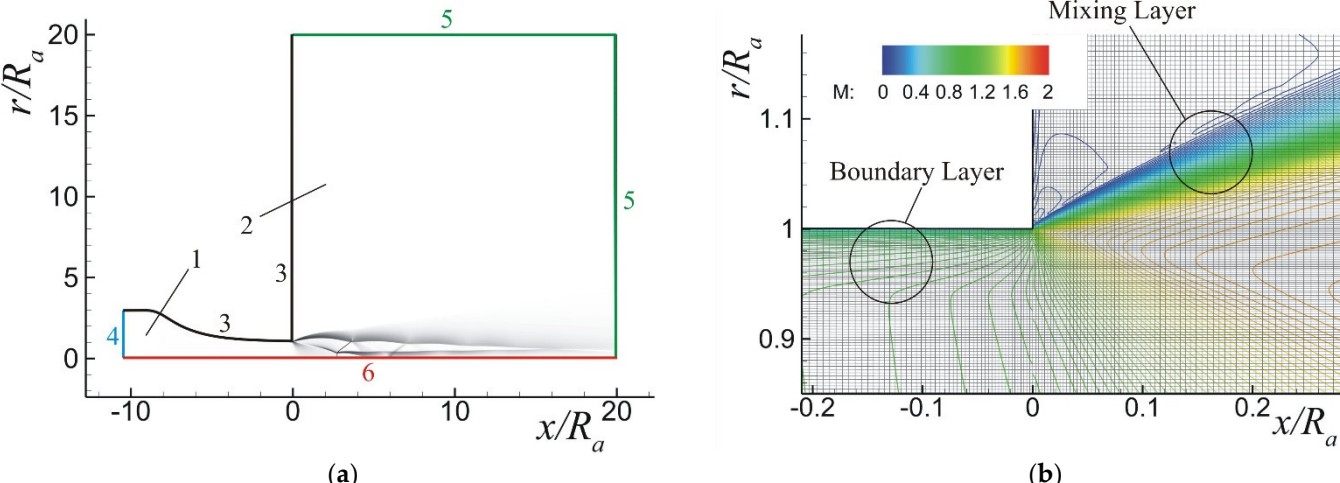

**Figure 4.** Computational domain (**a**): 1—nozzle, 2—ambient space, 3—wall, 4—inlet boundary, 5—outlet boundary, 6—axis of symmetry; (**b**) fragment of the grid near the nozzle exit.

Spatial discretization of the conservation and transport equations was performed using a first-order upwind scheme. Flux values were calculated using the Roe-FDS (flux-difference splitting) scheme [22]. Flow parameter gradients were calculated using the Green–Gauss cell-based method [23].

The choice of equations (RANS) and the scheme for their solution (first-order upwind) is due to the following considerations. In the case under consideration, the flow within the first "barrel" of the jet is of interest. The flow includes a shock-wave structure ("barrel" and reflected shocks and a Mach disk) and mixing layers—external and those coming from the shocks' internal interaction region. Of greatest interest is the shock-wave structure, which is primarily affected by the flow character at the nozzle exit, and, to a much lesser extent, turbulent processes in the mixing layers.

In this case, the adopted solution scheme in a two-dimensional formulation using the RANS turbulence model should not significantly affect the interesting features of the calculation results, and at the same time, it saves computational resources that can be directed to resolving the grid near the nozzle exit.

On the other hand, the application of such an approach leads to the appearance of hydrodynamic instability in the two-dimensional mixing layer, with large-scale vortices appearing near the nozzle exit, which are not provided by the chosen turbulence model. Therefore, in order to reduce the level of numerical fluctuations in the outer mixing layer, a scheme of the first order of accuracy was chosen. In addition, such a scheme makes it possible to remove non-physical oscillations of the flow parameters in the vicinity of shocks, which are caused by higher-order schemes, which simplifies the analysis of the flow in the vicinity of shocks.

The boundary conditions on the walls of the computational domain (Figure 4a), denoted by the number 3: the gas velocity is zero; there is no heat transfer between the nozzle surface and the gas. The calculation of the boundary layer inside the nozzle is carried out using the near-wall function [24,25]. The thickness of the boundary layer in the nozzle accounts for about 40 cells. The y+ parameter is 24, rising to 30 towards the nozzle exit.

Boundary conditions for solving the conservation equations (specified values): at the inlet (Figure 4a, indicated by number 4) pressure $p_0 = 0.44$ MPa, temperature $T_0 = 292$ K; at the outlet boundary (Figure 4a, denoted by number 5) pressure $p_e = p_c = 8.8 \times 10^{-2}$ MPa, temperature $T_e = 296$ K. Boundary conditions for the equations for calculating the turbulent parameters k and $\omega$ at the same boundaries (specified values): turbulence intensity $I = 0.05\%$; ratio of turbulent to molecular viscosity $\mu_t/\mu = 1$.

The flow parameters at these boundaries were calculated using the AUSM+ Liu scheme [26]. For a subsonic compressible flow (the Mach number that is normal to the boundary is less than 1), the parameters at the boundary are calculated as the weighted average values of the specified parameter and the parameter in the gas flow that is adjacent to the boundary. In the case of a supersonic flow that is normal to the boundary, the value of the parameter at the boundary is equal to the parameter in the flow that is adjacent to the boundary.

Boundary conditions on the axis (Figure 4a, denoted by the number 6): the radial velocity component and the radial gradients of the flow parameters are equal to zero.

The reliability of the calculation results was verified through their comparison with experimental data.

The mean Pitot pressure for the studied jet and the confidential interval measured in the control cross section $x/R_a = 2.0$ are shown in Figure 5 in black. The presented data are obtained from the results of measurements that were performed over a period of seven years. The largest pressure deviations are visible in the jet regions with a strong radial flow gradient, i.e., in the barrel shock and in the jet mixing layer.

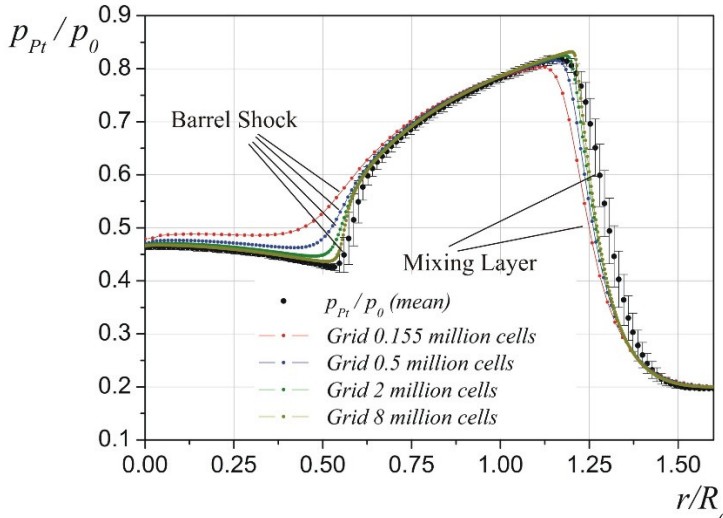

**Figure 5.** Comparison of calculation and experiment in cross section $x/R_a = 2.0$.

The numerical calculation was carried out on a sequence of refined grids, in which the size of cells along the x and r axes decreased by a factor of two each time. The solution on a grid containing two million cells is closest to the experiment, except for the intensity (pressure gradient in the radial direction) of the barrel shock and the thickness of the mixing layer. These features of the solution are caused by the influence of the first-order accuracy scheme and the chosen turbulence model, respectively. Despite these shortcomings, the obtained numerical results are suitable for the analysis and interpretation of experimental data.

## 4. Flow Structure at the Initial Section of the Supersonic Underexpanded Jet

Figure 6 shows the flow pattern in the first cell, or barrel, of the jet that is ejected out of the sonic nozzle with $M_a = 1$ and $n = 2.64$. Shown here are: (a) an experimental schlieren photograph; (b) visualization of numerical data. The photograph was taken with a Foucault knife in a horizontal position. This configuration is possible to observe by using the vertical gradient of air density.

The main elements of the jet are clearly seen, these being an expansion fan taking its origin at the nozzle edge; a barrel shock; a Mach disk; the boundary of the jet flow, or the outer mixing layer; and the slip line, or the mixing (shear) layer, forming behind the triple point.

The photograph shows a gradient flow feature G, which has an S-bending near the nozzle exit similar to that which is observed in the photographs of Figure 2. This feature looks like a barrel shock that has already formed directly near the nozzle exit. At the same time, the numerical visualization is more consistent with the diagram of Figure 1d, where the fan trailing edge is observed near the nozzle exit, and only later a barrel shock forms. This difference needs to be clarified.

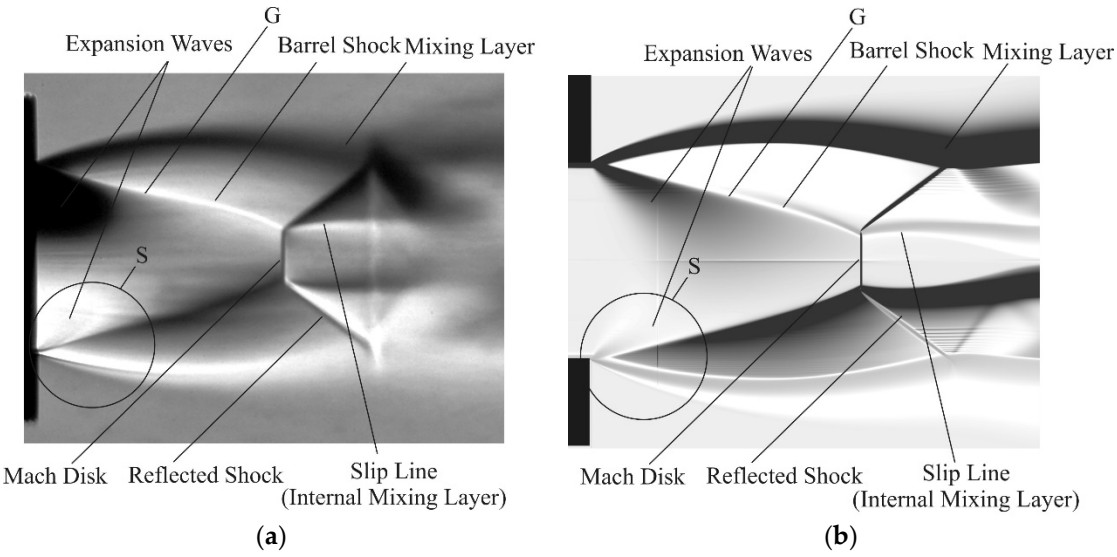

**Figure 6.** The first barrel of the jet ejected from the nozzle with $M_a = 1$ and n = 2.64: (**a**)—schlieren photograph taken at 2-ms exposure, (**b**) visualization of calculated data. Here G is the gradient flow feature, S is the region of its S-bending shape.

The position in space of the experimentally registered gradient flow non-uniformity G is shown in Figure 7. Here, a good correspondence between the data that are obtained using visualization patterns and the Pitot data is observed.

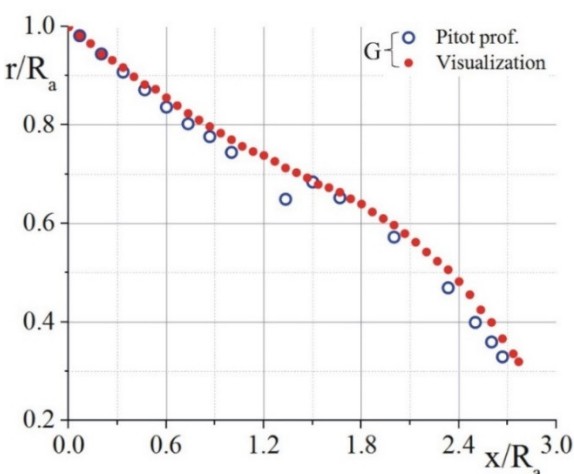

**Figure 7.** Spatial position of the gradient non-uniformity, or the line G in Figure 6, as determined from flow visualization and Pitot data.

For the measurements that were performed using a Pitot tube, the position of this non-uniformity was determined from the local minimum in the radial profiles of the relative total pressure $p_{Pt}(r)$ (see Figure 8, where this position is marked). Here, x is the axial coordinate, r is the radial coordinate, and $R_a$ is the radius of the nozzle exit section. The origin is located at the center of the nozzle outlet.

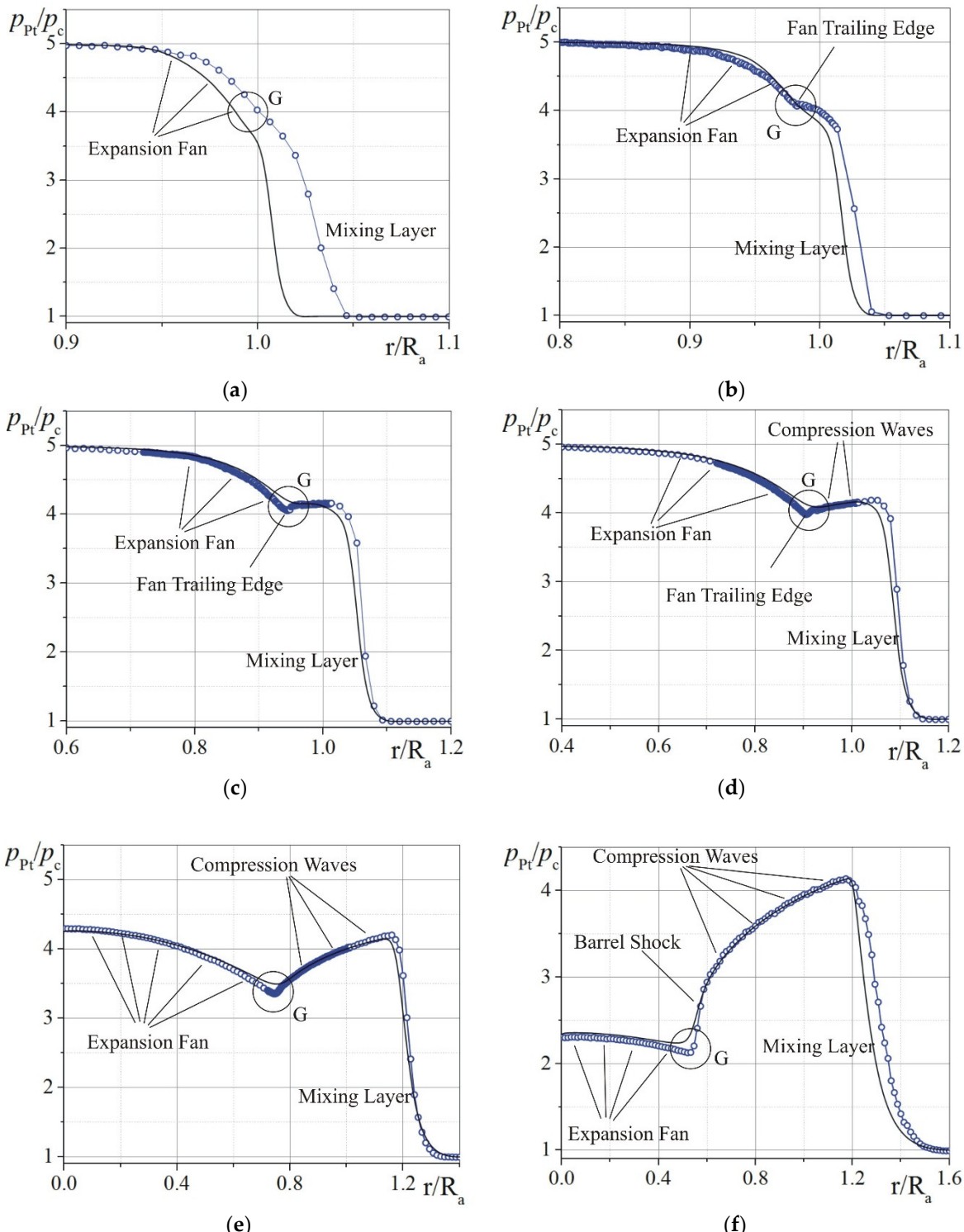

**Figure 8.** The radial distribution of the relative total pressure $p_{Pt}/p_c$ obtained in the cross-sections of the jet $x/R_a = 0.03$ (**a**), 0.07 (**b**), 0.2 (**c**), 0.33 (**d**), 1.0 (**e**), and 2.0 (**f**). The blue circles and black lines are, respectively, the experimental and numerical data.

Different slopes of the S-shaped curve corresponding to the position of the non-uniformity G in space with respect to the jet axis can be indicative of a different nature of the registered non-uniformity: near the nozzle exit, $x/R_a < 1.0$, this may be the trailing edge of the expansion fan, and at $x/R_a > 1.5$, a barrel shock. Such an interpretation does not correspond to the above analysis of the visualization results of Figure 6a, according to

which this feature is interpreted as a shock. In some cases, at this stage of the analysis of the results, it is concluded that the barrel shock is formed directly near the nozzle exit, and then the cause of its curvature is sought.

For example, it is assumed that the cause of the curvature may be an effect similar to that which is observed in the flow behind the aft cut of the cone [27,28]. The curvature of the shock near the nozzle exit is associated with the influence of the boundary layer on the inner wall of the nozzle, in which the fan of expansion waves near the point N(I), Figure 1c, is not centered. This interpretation does not contradict the interpretation of the sounding results (Figure 7) and requires further clarification.

A comparison between the experimental and numerical results is shown in Figure 8. Here, the radial profiles of Pitot pressure, or the measured total pressure that is normalized by the ambient pressure in the air surrounding the jet, $p_{Pt}/p_c$, in the sections $x/R_a = 0.03$ (a), 0.07 (b), 0.2 (c), 0.33 (d), 1.0 (e), and 2.0 (f) are shown. The Pitot pressure refers to the total pressure that is measured in the subsonic flow and to the total pressure behind the normal shock in the supersonic flow.

The black solid line shows the calculated data, and the blue hollow circles, experimental data. It can be seen from the figure that the results of the numerical calculation can be used to analyze and interpret the experimental data. Here, a mixing layer, an expansion fan, a fan of compression waves, and a barrel shock can be identified. The circle marks the gradient feature G of the flow, whose position in space is shown in Figure 7. According to the diagram of Figure 1d, in Figure 6b–d this flow feature G can be interpreted as the trailing edge of the expansion fan, separating out the expansion fan from the compression fan. In Figure 7f, this feature corresponds to a barrel shock. However, the previous interpretation of this feature as a shock can also be correct.

Below, a more detailed analysis of the obtained experimental results is carried out with the use of numerical calculation data.

Figure 9a shows the calculated distribution of the isolines of relative density $\rho/\rho_e$ in the first barrel of the jet (here, $\rho_e$ is the air density in the region surrounding the jet). Numbers 1–4 denote the cross sections $x/R_a = 0.067$, 0.47, 1.0, and 2.0. The line G is the gradient flow non-uniformity. Figure 9b shows the distribution of relative static pressure $p/p_e$ and relative gas density $\rho/\rho_e$ in the same cross-sections. The circles show the distribution of pressure, and the solid lines, that of air density. The following flow features are clearly seen: the jet boundary ($p \sim p_e$); the mixing layer (at the outer boundary of this layer, we have $\rho \sim \rho_e$ and $M \sim 0$); and the gradient flow non-uniformity G, in which the second derivative of the gas-dynamic parameters of the flow reaches an extreme value.

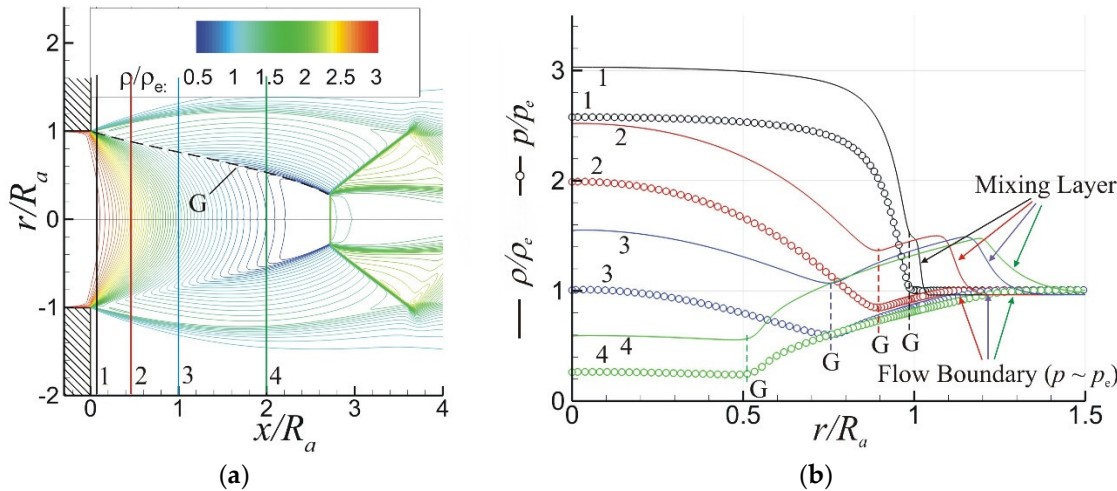

**(a)**            **(b)**

**Figure 9.** Distribution of Mach-number isolines (**a**) and profiles (**b**) of the relative values of static pressure $p/p_e$ and density $\rho/\rho_e$ in the sections $x/R_a = 0.07$ (1), 0.5 (2), 1.0 (3), and 2.0 (4) of the first barrel of the jet along the transverse coordinate $r/R_a$.

It is seen from the figure that the position of the line G in space can be identified both from the distribution of air density (in the experimental photographs, depending on the settings of the shadow device, the distribution of either the first or second derivative of air density can be registered), and from the distribution of pressure. This allows further analysis of the flow to be performed using numerical-calculation data for any distribution of flow gas-dynamic parameters without invoking experimental results.

The process of formation of the barrel shock can be traced, considering Figures 10 and 11. Figure 10a shows the distribution of Mach-number isolines in the region of the first jet barrel. The solid black lines show the characteristics in the supersonic flow region.

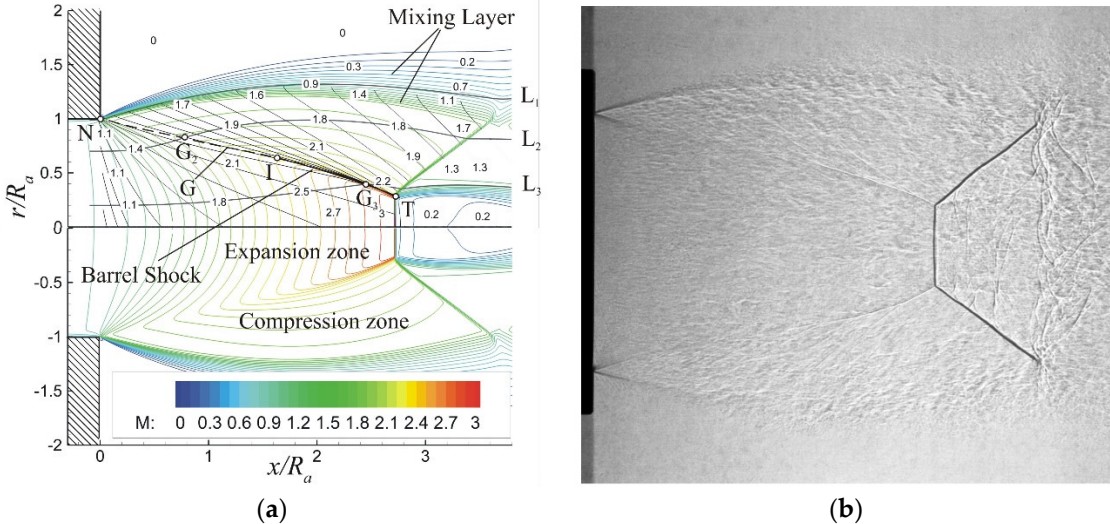

(a)  (b)

**Figure 10.** Diagram illustrating the formation process of the gradient flow feature G and that of the barrel shock (**a**), and the corresponding shadowgraph (**b**).

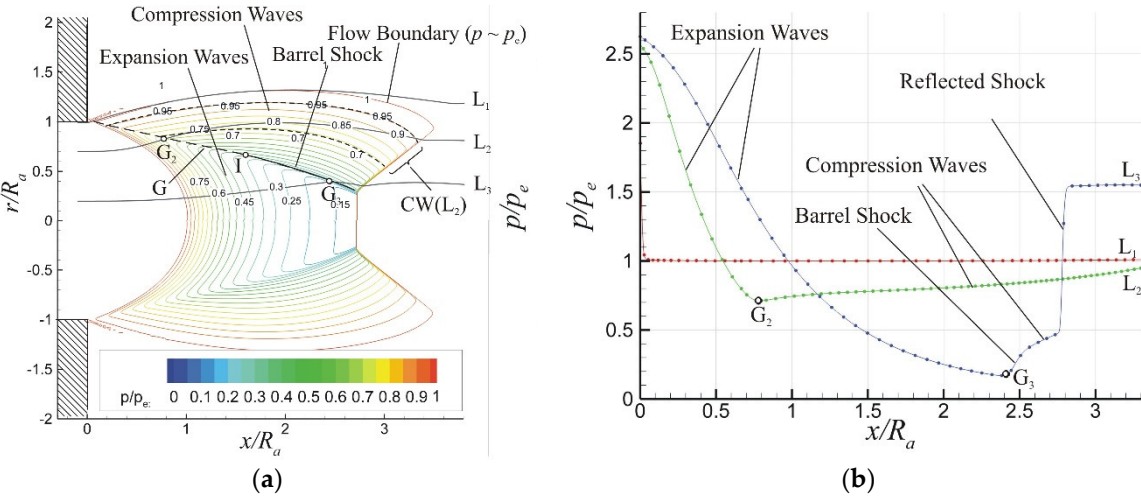

(a)  (b)

**Figure 11.** The passage of the streamlines through the region occupied by the compression waves: the distribution of pressure isolines in the regions of the first barrel (**a**) and the distribution of pressure along the streamlines (**b**).

A centered fan of expansion waves emanating from the point N and occupying the paraxial region is clearly seen. In the peripheral region, characteristics forming compression waves are observed. The first characteristic emerges from the point N, thus forming the line G. The characteristics following this line in a downstream direction, departing from the jet-flow boundary, are superimposed on it, forming a barrel shock. According to the numerical

calculation, the shock formation region (point I in the figure) is located at a distance from nozzle exit $x/R_a \sim 1.7$. A closer examination of an instantaneous shadowgraph picture taken at 4-μs exposure (see Figure 10b) shows that the barrel shock becomes visible starting from distances $x/R_a > 1.5$. Three characteristic streamlines $L_1$–$L_3$ are shown in gray color. The streamline $L_1$ descends directly from the nozzle wall, delineating the boundary of the jet flow that is ejected out of the nozzle. Located outside of this boundary is a part of the jet mixing layer formed by the gas that is ejected by the jet from the surrounding space. Located inside the boundary is the gas that is exhausted out from the nozzle. The streamline $L_2$ passes through the line G in the peripheral region of the jet (the streamline $L_2$ crosses the line G at the point $G_2$), and the streamline $L_3$ passes through the line G in the paraxial region, at the point where the barrel shock is formed (the streamline $L_3$ crosses the line G at the point $G_3$). It should be noted here that, according to the calculated data, when crossing the feature G, the streamlines $L_1$–$L_3$ behave differently: the streamline $L_2$ retains its direction, whereas the streamline $L_3$ abruptly changes its direction, this circumstance indicating the absence of a shock near the nozzle exit and the presence a shock near the Mach disk.

The CFD distributions of static pressure are shown in Figure 11.

Figure 11a shows the isolines of the relative static pressure $p/p_e$ near the nozzle exit in the first barrel of the jet. For the convenience of a flow analysis, the isolines in the interval of $p/p_e$-values ranging from 0 to 1 (up to the pressure in the region of the stationary gas surrounding the jet) are shown. According to [1], the isoline $p/p_e = 1$ is the boundary of the jet flow. Initially, the boundary streamline $L_1$ almost exactly follows the isoline $p/p_e = 1$, but then it departs from the latter isoline, moving into the outer region of the mixing layer. The latter indicates a gradual reduction in Mach number along this isoline from the nozzle exit in a downstream direction, due to the mixing of the jet gas with the gas from the surrounding space. The streamline $L_2$ passes through the gradient feature G and enters the compression region. It can be seen from the isolines that, as the flow moves along this region from the point $G_2$ to the reflected shock, the pressure $p/p_e$ increases in magnitude in the compression fan $CW(L_2)$ from 0.7 to 0.95. A similar increase in the pressure along the streamline $L_3$ between the barrel and reflected shocks amounts to 0.2 to 0.5 of $p/p_e$. The variation of pressure along these streamlines as a function of the distance to the nozzle exit is shown in Figure 11b. The main elements of the structure, such as the expansion and compression waves, and the barrel and reflected shocks, can be traced here.

The results that are presented above allow us to conclude that the gradient flow non-uniformity G that is observed in the experimental photographs in the form of a solid line near the nozzle exit is a characteristic (called in [12] the fan trailing edge) separating out the rarefaction and compression regions, and at some distance from the nozzle exit this feature transforms into a barrel shock. The performed analysis well agrees with the diagram of Figure 1d.

The reason for the S-bending of the gradient line G (like that in Figure 2) can be explained considering Figure 12, which shows the structure of an axisymmetric (upper half of the figure) and a flat two-dimensional (lower half of the figure) jet with $M_a = 1$ and $n = 2.64$. The density isolines are shown here in gray, and the characteristics, in black.

The picture of the two-dimensional planar flow in the vicinity of the nozzle exit is close to the Prandtl–Meyer flow. A centered expansion fan is formed at the nozzle exit (the characteristics of the fan emanate from the point N). Behind the expansion fan, there forms a quasi-regular flow region, in which the characteristics are roughly parallel to each other and the flow parameters along the characteristics themselves vary according to the drawn isolines. The characteristic separating out the rarefaction and quasi-uniform flow regions is designated as G(p). Unlike in the case of an axisymmetric jet, there is no shock near the G(p) line, and the characteristic itself exhibits no S-shaped bending.

At the top of the figure, the flow pattern of an axisymmetric jet is shown; the solid line G(a) shows the gradient feature of the axisymmetric jet, and the dotted line shows the gradient feature G(p) for the case of a flat jet.

Evidently, the transverse size of the axisymmetric jet is much smaller than that of the plane jet. This is because, in a plane jet, the flow expands in an expansion fan (similarly to the Prandtl–Meyer flow), whereas in an axisymmetric jet, an additional expansion of the flow occurs due to the increase in its azimuthal size (Figure 12 shows an explanatory diagram, located on the right). In this case, the pressure $p_e$ is attained earlier, causing a smaller transverse size (diameter) of the axisymmetric jet.

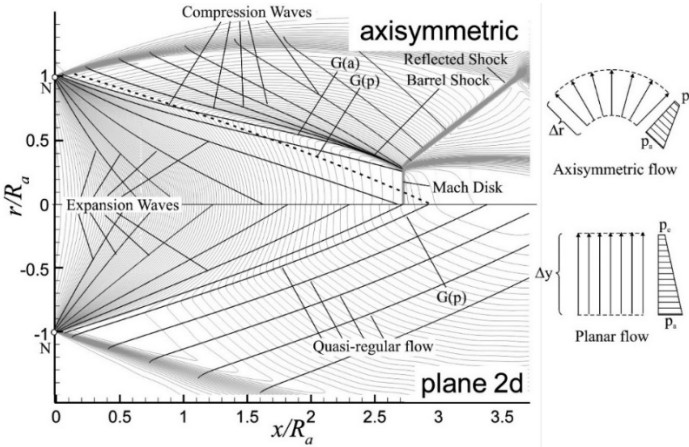

**Figure 12.** Comparison between the flow structure of the axisymmetric jet (**top**) and that of the plane sonic jet (**bottom**) for $M_a = 1$ and $n = 2.64$.

As a result of such a decrease in diameter, the characteristics extending from the jet boundary are superimposed onto the characteristic G separating out the rarefaction and compression regions with the formation of a barrel shock. In an axisymmetric flow, the intersection of the shock with the axis of symmetry leads to the formation of an irregular interaction with the appearance of a Mach disk and a reflected shock. Initially, near the nozzle exit, the lines G(a) and G(p) are close to each other, but further downstream at a distance $x/R_a > 1$ the line G(a) deviates towards the outer jet boundary, and this circumstance leads to its S-shaped bending.

It can be concluded that the reason for the observed S-shaped bending of the gradient flow feature G is the axisymmetric nature of the jet flow. Unlike in the case of the flat flow, here, the characteristics are reflected from the jet boundary and interact with the characteristic G separating out the rarefaction and compression regions, forming a barrel shock. The irregular interaction of the barrel shock with the axis of symmetry leads to the formation of a triple configuration, in which the barrel shock deviates from the jet axis, thus imparting the observed bending to this flow feature.

## 5. Conclusions

A detailed study of the gas-dynamic structure of the flow over the initial section of an underexpanded sonic jet exhausted into ambient space was performed. The structure of a gradient flow feature—sometimes interpreted as a barrel shock—taking its origin at the nozzle edge and reaching the region of the irregular shock-wave interaction, is analyzed.

It is shown that the interpretation of this feature, based only on the results of visualization and probe measurements, provides an erroneous conclusion about the formation of a shock that is directly at the exit edge of the nozzle.

The involvement of numerical calculation data showed that near the nozzle exit, this flow feature presented a characteristic separating out of the expansion and compression regions. Further downstream, this flow feature transforms into a barrel shock. The mechanism of formation of this flow feature has been proposed in [12].

As a result of the axisymmetric and plane supersonic jets patterns comparison, it was established that the observed S-shaped curvature of this gradient feature was caused by

the axisymmetric nature of the flow. In a plane jet, the flow expands in a fan of expansion waves (the flow is close to the Prandtl–Meyer flow) without the formation of a shock, and in the axisymmetric case, due to additional expansion of the flow in the azimuthal direction, the jet narrows with the formation of a barrel shock and its curvature due to the Mach interaction with the jet axis.

**Author Contributions:** V.Z. was leader and was responsible for the written paper; I.K. and N.K. were responsible for the written paper and performed the experiments. All authors have read and agreed to the published version of the manuscript.

**Funding:** The research was carried out within the state assignment of the Ministry of Science and Higher Education of the Russian Federation (project No. 121030500158-0). The study was conducted at the Equipment Sharing Center «Mechanics» of ITAM SB RAS.

**Institutional Review Board Statement:** Not applicable.

**Informed Consent Statement:** Not applicable.

**Data Availability Statement:** Not applicable.

**Conflicts of Interest:** The authors declare no conflict of interest.

## Nomenclature

| | |
|---|---|
| $N_{pr}$ | Nozzle pressure ratio |
| $n$ | The non-calculated pressure ratio = $P_a/P_c$ |
| $M_j$ | Jet Mach number |
| $M_a$ | Mach number on the nozzle exit |
| $p_a$ | Pressure at the nozzle exit |
| $p_c$ | Pressure in the ambient space |
| $p_0$ | Total pressure |
| $p_{Pt}$ | Pitot pressure |
| $x$ | Longitudinal coordinate |
| $y$ | Transverse coordinate |
| $Re_d$ | Reynolds number = $U_a \cdot D_a / \nu$ |
| $\nu$ | Kinematic viscosity |
| $T_0$ | Total temperature |
| $T_c$ | Static temperature |
| $I$ | Turbulent intensity |
| $\mu$ | Laminar viscosity |
| $\mu_t$ | Turbulent viscosity |

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
