# Peer review of "Flow Feature in Supersonic Non-Isobaric Jet near the Nozzle Edge"

_aerospace, doi:10.3390/aerospace9070379_

Round 1

Reviewer 1 Report

The article is well written and clear.

The description of the research topic in the introduction is clear with relevant references. The experimental part is well discribed. Could you precise the part of measurement error induced by the intrusivity of the probe into the compressible flow ? Is the optical imagery of the flow  performed with a shadow device providing the detection if the secondery derivative of refractive index (ie density) or schlieren system ? Could you clarify ? IAB-451 is classicaly a schlieren system ? After all,  the quality of the pictures is very good.

Concerning the numerical calculation, it is known that kw-SST model is a good choice for RANS calculation for under-expanded flows. But it could introduce inaccurate simulation of turbulence behavior of the flow leading to unconsistencies in the second part of the plume. Here you focus on the first barrel shock and comparaisons with experimental data provide very consistant results which may validate the numerical approach.   In the description of the calculation, I am not convinced that a first-order upwind scheme  could provide such results. Could you check your sentence ? Did you test a convergence of the mesh ? Did you use a refinement process in order to improve the accuracy of shock's calculations ?

Concerning part 4, the figure should be improved with error bars for both measurement systems.

The exploitation of numerical results in order to describe the formation of barrel shock is quite interesting and relevant.

Author Response

We thank reviewer for the positive feedback. All comments on the article text were useful and allowed to significantly improve the quality of the material presentation. In attachment there are the responses to the reviewer's comments indicating the changes made to the article text.
Sincerely, the authors team

Reviewer 2 Report

The article presents experimental observations and numerical calculations of a supersonic non-isobaric jet flow. The authors identify flow features based on the jet visualizations and the numerical data.

The paper is well written and indicates compelling observations of the flow features. However, the authors draw conclusions based on a 2D calculation using the RANS approach, which is not well suited for such flow configurations. Moreover, the numerical data diverges from the experimental observations presented in the article. The numerical results are merely qualitative and reinforce the experimental observations, but they do not provide enough accuracy to provide strong conclusions as the authors do in the paper.

The article abstract is brief and partially describes the work since it does not cite using numerical calculations to support the experimental observations of the jet flows. Moreover, it is crucial that the abstract mentions the use of the 2D RANS formulation.
The introduction mainly describes different experimental studies of the jet flow and its conclusions. However, it does not discuss the state of the art of numerical calculation of such flow configuration. I strongly suggest that the authors add a review on numerical studies of supersonic jet flows, and I provide a few citations that could improve the publication:

- Bodony, D. J., & Lele, S. K. (2008). Current status of jet noise predictions using large-eddy simulation. AIAA journal, 46(2), 364-380. (https://doi.org/10.2514/1.24475)

- Bodony, D. J., & Lele, S. K. (2005). On using large-eddy simulation for the prediction of noise from cold and heated turbulent jets. Physics of Fluids, 17(8), 085103. (https://doi.org/10.1063/1.2001689)

- Mendez, S., Shoeybi, M., Sharma, A., Ham, F. E., Lele, S. K., & Moin, P. (2012). Large-eddy simulations of perfectly expanded supersonic jets using an unstructured solver. AIAA journal, 50(5), 1103-1118. (https://doi.org/10.2514/1.J051211)

Chauhan, M., & Massa, L. (2022). Large-Eddy Simulation of Supersonic Jet Noise with Discontinuous Galerkin Methods. AIAA Journal, 60(3), 1451-1470. (https://doi.org/10.2514/1.J060424)

- Brès, G. A., Ham, F. E., Nichols, J. W., & Lele, S. K. (2017). Unstructured large-eddy simulations of supersonic jets. AIAA journal, 55(4), 1164-1184. (https://doi.org/10.2514/1.J055084)

Abreu, D.F., Junqueira-Junior, C., Dauricio, E.T.V., and Azevedo, J.L.F., “A Comparison of Low and High-Order Methods for the Simulation of Supersonic Jet Flows,” Proceedings of the 26th ABCM International Congress of Mechanical Engineering – COBEM 2021, Paper COB-2021-0388, ABCM, Rio de Janeiro, Virtual Congress, 22–26 November 2021 (10 pages; doi://10.26678/ABCM.COBEM2021.COB2021-0388).

Experimental fluid mechanics is far beyond my expertise. Therefore, I will get into the details of the experimental equipment section of the paper.

The numerical calculation section is of paramount concern. The authors describe the choice of 2D RANS formulation using a first-order accuracy spatial discretization. Using 2D calculations for such a complex flow configuration can drive erroneous conclusions since it assumes symmetric behavior of the flow by construction. Some flow features can be asymmetric, and this approach would mask such behavior. Moreover, the authors do not clarify if they are using a URANS or RANS formulation since they do not mention the nature of the time step. Hence, it is impossible to say if the numerical results are an averaged solution or the steady-state convergence of the averaged Naver-Stokes formulation. Such information plays a crucial role in turbulence transition and stability of the flow. Furthermore, the RANS type turbulence models present major weaknesses when studying unsteady compressible flows (Lele, S. K. (1994). Compressibility effects on turbulence. Annual review of fluid mechanics, 26(1), 211-254. https://doi.org/10.1146/annurev.fl.26.010194.001235). The computational fluid dynamic community applies large-eddy simulations for calculating the supersonic jet flow configurations

One should also mention that first-order upwind schemes are highly dissipative and could be the reason for the divergence of the results exhibited in Figure 7. Moreover, the calculations would be reproducible if the authors explicitly added a citation of the upwind scheme formulation to the text.

When discussing the computational domain and boundary conditions, I would start by suggesting the addition of an illustration of the computational grid to improve the article. Moreover, the authors present values imposed at the boundary conditions but do not explicitly indicate what type of boundary condition they apply. Is it a zeroth-order extrapolation for the supersonic inlet? Do the authors impose an inviscid boundary layer profile at the entry of the nozzle? Are Riemann invariants imposed in the far-field region? A description of the boundary condition is mandatory to fully express the numerical formulation.

Numerical formulation based on a simplified hypothesis, such as the RANS formulation, can provide only qualitative support for a physical behavior captured with experimental techniques. The authors can say that their calculations provide more arguments for their observations. However, they cannot draw any conclusions based on their data. A more accurate simulation with sophisticated post-processing techniques is necessary for this purpose.

The authors should modify the last phrase of section 3, page 5, line 149: "The reliability of calculation results was checked through their comparison with experimental data", to a more precise phrase indicating that the numerical calculations provide a qualitative description of the flow and could be used to support the experimental data.

Section 4 "Flow structure at the initial section of the supersonic underexpanded jet" presents the results and discussions of the article. The results are interesting, but, once again, the authors indicate the numerical data as a solid foundation for building their conclusions and observations, which is not the case. Figures 7 (a) and (b) indicate significant divergences between numerical and experimental data when evaluating pressure profiles. One can say the differences are proof that the simulation presents weakness when predicting such flow configuration.

The authors should again modify the phrase on page 8, line 192 when they say: "A satisfactory agreement between the two datasets is evident". The statement should be more realistic and indicates that the numerical simulations present difficulties in precisely representing the positioning of shock waves and discontinuities but it can be used as qualitative support for experimental observations.

On page 10, line 267, the authors indicate, once again, that the comparisons allow the conclusion on the formation of barrel shocks. This expression requires modification indicating that the current work presents evidence of the barrel shock formation, and more refined numerical studies could lead to a final conclusion.

The article presents interesting experimental observations of the underexpanded jet flow. The qualitative calculation results corroborate such observations. The authors should only improve the discussion by mentioning the limitations of the numerical approach used in the present work and the need for complementary simulations using more sophisticated approaches to conclude their observation.

Author Response

(The authors gave the same response as above.)

Reviewer 3 Report

The authors investigate the barrel shock structure in the first shock cell of a supersonic jet. The authors utilise numerical simulations and experiments to analyse the flow.

Would be nice if the autors could add a paragraph motivating the study. Why is this research needed? What is the aim?

The authors call the turbulence model the "two-parameter SST differential k-ω model". Do you mean the standard k-omega SST model?

The viscosity ratio is a boundary condition for external flows. The turbulent length scale is the right boundary condition for internal flows.

Please be more explicit with the quantity that is shown in figure 5 b. Numerical data is not precise. The reader would like to know if the images show the same quantity or not and if they can be compared or not.

Certainly, the images in figure 5 show some differences close to the nozzle. The expansions start right at the nozzle exit for the experimental visualisation while they start further downstream in the numerical simulations. Why?

What are the numerical artefacts manifesting as horizontal lines in the proximity of shocks?

Why is there a radial gradient change of the boundary layer flow in the nozzle?

Please provide the maximal y+ values so that one can get an idea of the mesh size.

There is an extra dot in line 237. In line 253, please add a space after the dot.

Please separate the acknowledgement from the conclusions.

I feel that the word "shadow photographs" is slightly misleading and should be replaced by "shadowgraphs".

There are very few references in the manuscript. Please add some further references. E.g. there are no references to numerical methods. For example using a similar numerical method:

  • Semlitsch and Mihaescu (2018) "Fluidic Injection Scenarios for Shock Pattern Manipulation in Exhausts"
  • Gustafsson et al. (2012) "Nozzle throat optimization for supersonic jet noise reduction"

Author Response

(The authors gave the same response as above.)

Round 2

Reviewer 2 Report

I have only one comment regarding Figure 5 on page 7 of 15.

The picture describes the mesh size of billions of points while the text indicates that the most refined grid uses 2 million points. Considering the use of a RANS approach, I believe that the text is correct and the information divergence must be reviewed.

Best regards

Author Response

The authors thank the reviewer. In accordance with the remark, Figure 5 has been modified.
Sincerely, authors team

Reviewer 3 Report

The authors answered all my questions and comments.

Author Response

The authors thank the reviewer. In accordance with the remark the article has been modified.
Sincerely, authors team.